# Echinocandins as Biotechnological Tools for Treating *Candida auris* Infections

**DOI:** 10.3390/jof6030185

**Published:** 2020-09-22

**Authors:** Elizabete de Souza Cândido, Flávia Affonseca, Marlon Henrique Cardoso, Octavio Luiz Franco

**Affiliations:** 1S-Inova Biotech, Programa de Pós Graduação em Biotecnologia, Universidade Católica Dom Bosco, Campo Grande 79117900; Brazil; betty.souza@gmail.com (E.d.S.C.); flavia.affonseca27@gmail.com (F.A.); marlonhenrique6@gmail.com (M.H.C.); 2Centro de Análises Proteômicas e Bioquímicas, Universidade Católica de Brasília, Brasília 70790160, Brazil

**Keywords:** lipopeptides, echinocandins, *Candida auris*, infections, antifungal drugs

## Abstract

*Candida auris* has been reported in the past few years as an invasive fungal pathogen of high interest. Its recent emergence in healthcare-associated infections triggered the efforts of researchers worldwide, seeking additional alternatives to the use of traditional antifungals such as azoles. Lipopeptides, specially the echinocandins, have been reported as an effective approach to control pathogenic fungi. However, despite its efficiency against *C. auris*, some isolates presented echinocandin resistance. Thus, therapies focused on echinocandins’ synergism with other antifungal drugs were widely explored, representing a novel possibility for the treatment of *C. auris* infections.

## 1. Introduction

Candidiasis is one of the most common causes of fungal infection on a global scale and includes both superficial and invasive infections. The major concern is associated with patients in intensive care units (ICU) with high mortality rates. There are several fungal species isolated in the clinical setting responsible for these infections. *Candida albicans* is the most studied and frequently isolated species from nosocomial infections, but recently a new species, named *Candida auris,* has raised great concern regarding disorders caused by fungi [1,2].

*C. auris* is a multidrug-resistant pathogen that has been identified in 39 countries and has spread across five continents, with a mortality rate of ~78% [3]. Furthermore, risk factors that aggravate *C. auris* infections include conditions such as diabetes mellitus, cardiovascular surgical interventions and gastrointestinal pathologies, hematological malignancies and even corticosteroid therapy [4]. In addition, this pathogen spreads easily in hospital environments, but its correct identification is usually challenging. This new species appears as a major concern for health systems, since several isolates have already been confirmed as resistant to three antifungals drugs classes [5,6]. *C. auris* was reported for the first time in 2009. The strains were isolated from the external auditory canal of a Japanese patient. This fungus was discovered through the sequencing of the ribosomal DNA D1/D2 domain and nuclear ITS region. It was reported that *C. auris* is phylogenetically close to *Candida ruelliae* and *Candida haemulonii* [7]. In the same year (2009), 15 patients with chronic otitis media presented *Candida* species very closely related to *C. haemulonii* in South Korean hospitals [8].

Over the years, numerous other countries have isolated *C. auris* strains (Figure 1), including Kuwait [9,10], South Africa [11], Venezuela [12], Spain [13], Oman [14], Israel [15], Colombia [16], Panama [17], Pakistan [5], the United Kingdom [18], Australia [19] and Saudi Arabia [20], among others.

Currently, it is possible to separate *C. auris* isolates into five different clades, according to their geographical origin (South Asian, East Asian, South African, South American and Iranian). This classification was carried out using whole genome sequence data from global clinical data. Each of these clades differs phenotypically. Moreover, in terms of virulence, these isolates follow the order: South American > South Asian > (South African = East Asian) [4,21].

The emergence and exponential increase of nosocomial infections by *C. auris* leads to a need to seek new treatment options and infection control approaches [22]. A promising drug family that meets these criteria is the lipopeptides. These compounds are formed by a specific lipophilic fraction linked to an anionic peptide (six to seven amino acids). Lipopeptides have been shown to have antibacterial, antifungal, anti-adhesion and anticancer activities [23]. The major concern about *C. auris* infections is the fact that the vast majority of isolates already identified and evaluated have resistance to the four main antifungals classes: polyenes, azoles, echinocandins and nucleoside analogues, complicating the treatment of affected patients and, consequently, increasing the mortality rates [22]. Multiresistant strains bring major limitations to antifungal treatment options for patients affected by candemia, mainly due to the reduced health status of many of these patients. To date, several resistance mechanisms presented by *C. auris* against known antifungals have been studied. For example, specific mutations in the cellular target of antifungals, drug target overexpression, efflux pumps and biofilm formation are well-described resistance mechanisms [24].

Bearing this in mind, this review aims to provide a comprehensive update of the worldwide picture of *C. auris* infections, as well as presenting new views on the potential use of lipopeptides, especially echinocandins, as biotechnological tools for treating multiresistant *C. auris* nosocomial infections.

## 2. Echinocandins as Tools for Treating *Candida auris* Infections

Despite the increasing resistance presented by *C. auris* strains, lipopeptides remain the most prominent drugs among the current antifungal therapies. Biosurfactant molecules, such as lipopeptides, are particularly interesting as antifungals due to their high activity on the cell surface and high antibiotic potential. The absorption of biosurfactant molecules on the cell surface is related to changes in its hydrophobicity, which consequently causes changes in the fungus adhesion processes. Moreover, it is known that the hydrophobicity of the cell surface is related to adhesion and to some pathological processes in fungi, including *C. albicans* [25]. Lipopeptides are capable of self-assembling to form peptide-functionalized supramolecular nanostructures. Additionally, these amphiphilic molecules can incorporate one or more lipid chains attached to a peptide-head group, which makes these molecules advantageous for therapies, as this property facilitates the presentation of the peptide portion on the surface of nanostructures such as fibrils, micelles and vesicles [26].

Most naturally expressed lipopeptides have a cyclic hexapeptide head attached to a single lipid chain, responsible for their antifungal activity [27]. Thus, the cyclization of the peptide unit that occurs in many lipopeptides that are expressed by bacteria increases in vivo stability compared to their linear counterparts. This higher stability is due to reduced proteolysis resulting from the protection of C- and N-terminals. It also interferes with the molecule′s flexibility, but it seems to be related to the activity as well [26].

The echinocandins are lipopeptides naturally expressed by fungi such as *Glarea lozoyensis* [28], *Colephoma empetri* [29] and *Aspergillus nidulans* [28] among others [27]. These molecules are capable of blocking specific enzymatic reactions in the synthesis of cell wall essential components, the insoluble polysaccharide component acting at the β-1,3-glucan synthase and chitin, as observed in Figure 2 [25,30]. In *Candida* species, the FKS genes have been reported as essential in the catalysis of glucan synthase subunits. Glucan synthase represents a multienzyme complex, which comprises an integral catalytic membrane protein (FKS) and a regulatory subunit RHO1 protein. The latter is considered as a possible glucan synthesis activator. Thus, mutations in such genes are important in fungal resistance to echinocandins [31,32]. Echinocandins are recognized for being the first line in treating *C. auris* infections, since just 5% of the isolates are resistant to this drug class [33,34,35].

Echinocandin-type molecules, such as echinocandins FR901379 and WF11899A, B and C, are well studied and widely known as potent antifungals, especially against yeasts [36]. These molecules are particularly indicated for the treatment of invasive infections caused by *Candida*, especially in hemodynamically unstable patients after triazoles treatment. [27].

In the mid-1970s, echinocandin B was discovered, followed by the precursors of caspofungin (1989) and micafungin (1990) [31,37]. Together, these three molecules have been widely used as prophylaxis in combating *Candida* species infections. Echinocandins are not easily absorbed by the organism during oral dosage, making intravenous administration the best treatment option [38]. Despite the great number of molecules (Table 1) that have been studied in vitro and in vivo, currently only three echinocandins are approved by the FDA (The Food and Drug Administration) for use in the treatment of fungal infections, anidulafungin, caspofungin and micafungin.

Anidulafungin is a semi-synthetic derivative of echinocandin B, which originated as a fermentation product of *A. nidulans*. Structurally, it is a 1-[(4R,5R)-4,5-dihydroxy-N2-[[4”-(pentyloxy) [1,1′,4′,1”-terphenyl]-4-yl] cabonyl] L-ornithine] echinocandin B. It is commonly used in the treatment of esophageal candidiasis, candidemia and deep organ infections. Although its effectiveness and safety studies remain unclear, the FDA approved this antifungal in February 2006 [37]. There is no consistency in the data related to the antifungal activity of echinocandins against *C. auris*, which is understandable, as new isolates appear constantly, and not all of them present mutations in the FKS genes (which until now has been shown to be a solid condition in determining resistance to echinocandins). Therefore, recent studies have shown that some isolates of *C. auris* are tolerant to this molecule, and its fungicidal activity against some strains is reduced with MICs ≥ 4 μg·mL^−1^ or even not able to reach fungicidal activity [1,5,39]. A recent work carried out by Romera and co-workers [40] also demonstrated a substantial increase in *C. auris* resistance to biofilm formation. Free-floating and sessile cells from *C. auris* were tested against amphotericin B, anidulafungin, caspofungin, fluconazole and voriconazole. Sessile biofilm cells increased the tested antifungals resistance [40].

Caspofungin is a derivative of the hexapeptide expressed by *G. lozoyensis*, which was modified by the addition of N-acylated fatty acid chain. Structurally, it is a 1-[(4R, 5S)-5-[(2-aminoethyl) amino] 15-N2-(10,12-dimethyl-1-oxotetradecyl)-4-hydroxy-L-ornithine]-5-[(3R)-3-hydroxy-L-ornithine] pneumocandin B0 diacetate [37]. The FDA approved caspofungin in January 2001 for use in fungal infections in adults; later in July 2008 it was also approved for use in children over 3 months old [37]. Currently, the use of caspofungin is indicated in cases of neutropenic patients who present high fever and suspected fungal infections. In addition, it can be used to treat esophageal candidiasis, peritonitis, intra-abdominal abscesses, and oral cavity infections caused by *Candida* [41]. As for anidulafungin, caspofungin has also presented some *C. auris* strains with reduced susceptibility, with MICs ≥ 2 μg·mL^−1^. Alarming data from Indian hospitals indicate a 37% rate of resistance to caspofungin, based on the analysis of 102 *C. auris* isolates [42]. Even so, experiments using animal models indicate these drugs as the most effective against *C. auris* infections, leading to the need for more tests [1,5,39].

Caspofungins are considered effective against yeasts that form biofilms. Echinocandins, in general, manage to destabilize the integrity of the cell wall and reduce its stiffness, consequently causing cell lysis due to low osmotic pressure. [27]. Notoriously, caspofungins are considered effective against yeasts that form biofilms. However, they are inactive against *C. auris* biofilms, an unexpected event, since these molecules are normally effective against *Candida* species biofilms [43,44]. In addition, they are not used as a treatment for urinary infections caused by *C. auris*, as they fail to reach the required therapeutic concentrations of these compounds in the urine [27]. The survival *C. auris* capability in hospital environments may be related to yeast biofilm formation. In this way, Sherry and co-workers [44] tested the *C. auris* ability to form biofilms and further demonstrated that the species can adhere to polymeric surfaces. In addition, their results demonstrated a significant increase in resistance, highlighting that caspofungin, usually effective against *Candida* biofilms, was ineffective against planktonic cells and *C. auris* biofilms [44].

Similar to caspofungin, micafungin originated from the cleavage and addition of an N-acylated side chain to the natural hexapeptide derived from *C. empetri*. Structurally, it is identified as 1-[(4R,5R)-4,5-dihydroxy-N2-[4-[5-[4-(pentyloxy) phenyl]-3-24 isoxazolyl] benzoyl]-l-ornithine]-4-[(4S)-4-hydroxy-4-[4-hydroxy-3-(sulfooxy) phenyl]-25 l-threonine] monosodium salt [27]. The FDA approved this molecule in March 2005 for use in adults and in 2013 for pediatric treatment. Its use covers the treatment of adult and child patients with esophageal candidiasis and more delicate cases such as hematopoietic cell translation during neutropenia, being considered effective in 86% of the cases [41]. Similarly to the two aforementioned echinocandins, there are already strains of *C. auris* resistant to micafungin with MICs ≥ 4 μg·mL^−1^ [1,5,35].

In addition to the echinocandins already discussed, rezafungin (also named CD101), a new drug in this class has been developed. It represents a structural anidulafungin analogue and has a hexapeptide with a lipophilic tail and choline moiety at the C5 ornithine position. Modification in this analogue were carried out in order to increase the plasma stability, aqueous solutions and at high temperatures, to avoid hepatotoxicity [45], as well as increasing its half-life (30 to 40 h) and safety profile [46]. According to studies by Berkow and Lokart [47], rezafungin presented better MIC results than did other echinocandins, ranging from 0.03 to 8 μg·mL^−1^, while towards the same isolates the values presented by micafungin (0.5 to >8 μg·mL^−1^), caspofungin (0.5 to >16 μg·mL^−1^) and anidulafungin (1 to >16 μg·mL^−1^) were lower. Lepak and co-workers [48] demonstrated the potent in vivo rezafungin activity against clinical strains of *C. auris*. The authors suggested that the clinical dose of 400 mg administered once a week is enough to meet or exceed the pharmacodynamics target for >90% of isolates, being a really useful drug for patients infected with *C. auris* [48]. Still, some isolates have already been described as rezafungin resistant [45].

Although still in small proportions, *C. auris* resistance to lipopeptides is of great concern to researchers. It is also important to note that the resistance of *Candida* species to echinocandins is uncommon, but has been growing, mainly due to its widespread prophylaxis, being detected predominantly in isolates that already show resistance to azoles, suggesting a possible cross-resistance event [49]. The Infectious Diseases Society of America (IDSA) recommends that azole susceptibility tests should be performed for all blood streams and all possibly relevant *Candida* isolates. As well as susceptibility test to echinocandins should be carried out mainly in cases of previous infections by *Candida glabrata* and *Candida parapsilosis* [50]. The occurrence of echinocandins resistance in *Candida* species was first reported in 2005, where strains with mutations in the *FKS* genes of resistant *C. albicans* (*FKS1*) and *C. glabrata* (*FKS2*) presented low sensitivity to caspofungin [27]. Recent reports indicate that ~2% of *C. auris* could be resistant to echinocandins [2,27]. *C. auris* tolerance to echinocandins was also in vitro reported, revealing that some isolates showed high tolerance to anidulafungin, micafungin and caspofungin [35,39].

## 3. Alternative Therapies and Clinical Applicability

Despite the great efforts of researchers around the world in the search for more effective therapies aiming at the treatment of candidiasis, studies specifically targeting the control of *C. auris* infections are still in their early stages. Thus, there is a lack of antifungal therapies that have already been tested against this species when compared to other *Candida* species. Still, it is possible to list interesting works focused on combining drugs for better results in the control of *C. auris* infections.

The application of synergistic therapies can represent success in the absence of efficient treatments. For example, the synergism between micafungin and voriconazole showed positive results against multiple *C. auris* strains. On the other hand, combinations of micafungin and fluconazole, as well as caspofungin and voriconazole and fluconazole showed indifferent synergistic results [51,52].

Mahmoudi and colleagues [53] analyzed the synergistic effects of caspofungin and anidulafungin in combination with geldanamycin. Geldanamycin is a benzoquinone that inhibits the chaperone Hsp90 ATPase activity. This drug is originally an anti-tumor agent produced by *Streptomyces hygroscopicus* var. *geldanus* [54]. Authors reported that indifferent results were observed for the combinations tested against *C. auris* strains. Similarly, combinations of azoles with geldamycin were indifferent against the same strains [53]. Furthermore, Nagy and colleagues [55] reported that combinations of farnesol and the three echinocandins approved by the F.D.A. showed positive results against biofilms of *C. auris*. Farnesol is a sesquiterpene alcohol quorum-sensing molecule, which works to prevent the formation of *C. albicans* biofilms. This drug works by inhibiting the support cascade Ras1-cAMP-PKA that acts at the hypha-to-yeast transition [55,56,57]. The synergism mechanism of these drugs is not completely understood; however, farnesol is believed to modulate the expression of genes linked to ergosterol biosynthesis [58]. Thus, deleterious effects on the cell wall caused by echinocandins added to the imbalance in the ergosterol synthesis modulated by farnesol, can act simultaneously, causing fungal cell wall and membrane damage [55]. Besides, farnesol can induce the production of reactive oxygen species and disruption of mithocondrial functions, which can lead to enhanced echinocandin activity [58]. The use of effective farnesol concentrations can present emerging toxicity [59]. However, they can be used as an adjuvant in the action of echinocandins in therapies of local action (lock therapy). For biofilm treatment, this strategy avoids the toxic effect, since farnesol acts only within a catheter [55,57,60].

Shaban and coworkers [61] showed additive effects in the synergism of caspofungin and carvacrol against *C. albicans* and *C. auris* isolates, significantly reducing both drug concentrations. Carvacrol, or cymophenol, is a phenolic monoterpene and is present in the essential oil of oregano (*Origanum vulgare*). The authors reported that carvacrol is capable of inhibiting fungal adherence. A possible mechanism for that is blocking or even interfering in the synthesis of adhesins, which are responsible for fixing the fungus on surfaces.

Bidaud and colleagues [33] recently demonstrated the effectiveness of combining the antimicrobial peptide colistin with caspofungin against *C. auris*. The authors described the synergistic activity of these two molecules, even against azole-resistant isolates. It is interesting to note that colistin alone does not have antifungal activity; however, it is suggested that echinocandins cause a change in the cell wall, facilitating the access of colistin to the fungal cell membrane. However, when tested in association with caspofungin, no synergistic effect was observed [33]. Despite the encouraging results with caspofungin, the difference observed in the behavior of these two compounds is still unclear [33,51]. It is worth noting that positive results of synergism of echinocandins and colistin were previously reported against *C. albicans*. Both caspofungin and aminocandin showed synergism from the concentration of 0.4 mg·L^−1^ of colistin and from 3.1 μg·mL^−1^ of the echinocandins tested [62]. These results together prove the potential of combined therapies of echinocandins and antimicrobial peptides, even in in vitro trials.

Antifungal therapies based on the use of echinocandins in general have been shown to be very effective and with few side effects for patients. Treatments with micafungin and caspofungin, intravenously administered, represent a low cost when compared to conventional drugs such as amphotericin B. Also, when dealing specifically with the control of infections caused by *C. auris*, the use of combined therapies proved to be more promising than the use of echinocandins alone. However, it is necessary to be attentive due to prolonged use of these molecules may trigger kidney problems (especially when combined with drugs as amphotericin B deoxycholate) and even resistance, as previously discussed. Furthermore, it is important to note that the tests regarding combination therapies were performed in in vitro assays, and that it is essential to carry out in vivo tests, considering toxicity analyzes and adequate concentrations for therapeutic application, mainly for patients in critical health conditions.

## Figures and Tables

**Figure 1 jof-06-00185-f001:**
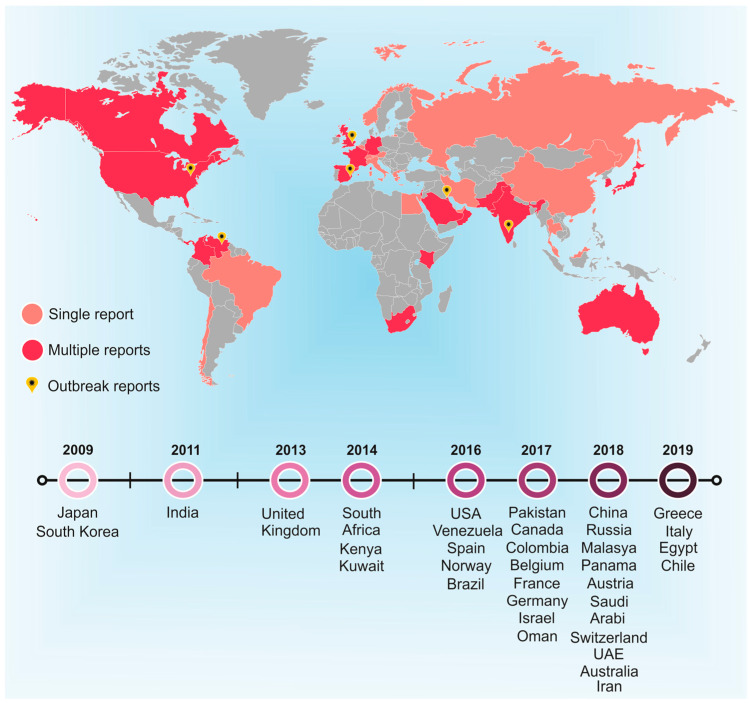
Global distribution of *Candida auris* between 2009 and 2019. Single cases are represented as (
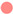
), whereas multiple cases are represented as (
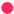
). (
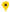
) Represents outbreak reports. The timeline presents the first report in each country highlighted over the years, with 2009 being the year in which the first case was reported.

**Figure 2 jof-06-00185-f002:**
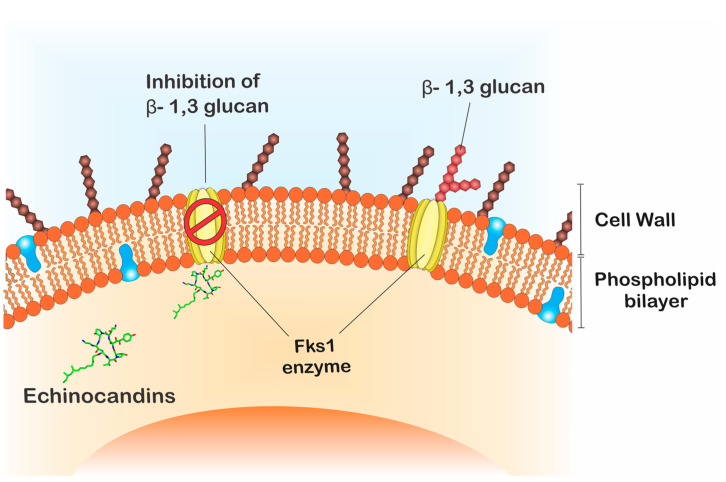
Echinocandins’ mode of action. Fks1 is essential for the catalysis of β-1,3-glucan, an important component of the fungal cell wall. Echinocandins acts through the inhibition of β-1,3-glucan synthesis.

**Table 1 jof-06-00185-t001:** Echinocandin entries in PubChem *.

Name	PubChem CID	Molecular Formula	Molecular Weight (g/mol)
1-((4R,5R)-4,5-dihydroy-L-ornithine) Echinocandin B	134693052	C_34_H_51_N_7_O_15_	797.8
1029890-89-8	134693051	C_34_H_52_CIN_7_O_15_	843.3
143131-16-2 Echinocandin B	456858	C_53_H_88_N_8_O_16_	1093.3
166663-25-8	15224271	C_58_H_73_N_7_O_17_	1140.2
79404-91-4	71762	C_49_H_71_N_7_O_17_	1030.099
79411-15-7	91632900	C_34_H_51_N_7_O_15_	797.8
Aminocandin	160772305	C_56_H_79_N_9_O_13_	1086.3
Anidulafungin	166548	C_58_H_73_N_7_O_17_	1140.2
Biafungin	92135635	C_63_H_85_N_8_O_17_ ^+^	1226.4
Caspofungin	2826718	C_52_H_88_N_10_O_15_	1093.3
CHEBI:2450	53297328	C_51_H_82_N_8_O_17_	1079.2
Cilofungin and Amphotericin B (AmB)	6473895	C_96_H_144_N_8_O_34_	1954.2
CINH3EtCOO Echinocandin	11984605	C_53_H_86_CIN_9_O_18_	1172.8
CINH3MeCOO Echinocandin	11984607	C_52_H_84_CIN_9_O_18_	1158.7
DiMeNEtOCOO Echinocandin	456505	C_55_H_89_N_9_O_19_	1180.3
Echinocandin B	9898144	C_52_H_81_N_7_O_16_	1060.2
Echinocandin B Nucleus	91820167	C_34_H_52_N_7_O_15_ ^+^	798.8
Echinocandin B Nucleus Hydrochloride	138115264	C_34_H_52_CIN_7_O_15_	834.3
Echinocandin C	10260509	C_52_H_81_N_7_O_15_	1044.2
Echinocandin D	12773979	C_52_H_81_N_7_O_13_	1012.2
Echinocandin Phosphate	23715870	C_50_H_80_N_8_NaO_20_P	1167.2
HOOCEtNHCOO Echinocandin	456500	C_54_H_85_N_9_O_20_	1180.3
HOOCMeNHCOO Echinocandin	456501	C_53_H_83_N_9_O_20_	1166.3
HOOCMeNMeCOO Echinocandin	456502	C_54_H_85_N_9_O_20_	1180.3
HOOCPrCOO Echinocandin	456504	C_55_H_86_N_8_O_20_	1179.3
L 731373	462493	C_50_H_82_N_8_O_16_	1051.2
Lipopeptide Der A-2a	456855	C_50_H_79_N_8_Na_2_O_18_	1157.2
Micafungin	477468	C_56_H_71_N_9_O_23_S	1270.3
Mulundocandin	121225706	C_48_H_77_N_7_O_16_	1008.2
Pneumocandin B0	5742645	C_50_H_80_N_8_O_17_	1065.2
Rezafungin	78318119	C_63_H_85_N_8_O_17_ ^+^	1226.4
Tetrahydroechinocandin B	171361	C_52_H_85_N_7_O_16_	1064.3
YKPHLXGEPNYRPY-UHFFFAOYSA-N	122233	C_50_H_81_N_7_O_16_	1036.2

* Duplicated entries were omitted. ^+^ Oxygen atom with more protons than electrons.

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
