# Peer review of "Echinocandins as Biotechnological Tools for Treating Candida auris Infections"

_jof, 2020, doi:10.3390/jof6030185_

Round 1

Reviewer 1 Report

  1. Line 29. It is not true that most C. auris isolates are resistant to the main 3 antifungal classes.
  2. Line 33. Chronic otitis media or externa?
  3. Lines 46-47: Again, the Authors considered C. auris resistant to all antifungal classes.
  4. Lines 54-55. Although, the Authors wanted to provide a comprehensive update of C. auris infections, they did not mention:

-the five different clades of C. auris with different phenotypic characteristics, virulence and antifungal susceptibility profiles (Szekely et al., J Clin Microbiol. 2019;57(5). pii: e02055-18, Forgács et al., Emerg Microbes Infect. 2020 Dec;9(1):1160-1169. doi: 10.1080/22221751.2020.1771218.).

-risk factors for C. auris infections.

  1. Line 88. “in fragile patients”. What does it mean? Probably, critically ill patients.
  2. Line 102. “deep-tissue candidiasis”. What does it mean?
  3. Line 119. “cavity infections”. What does it mean?
  4. Lines 124-133. “Caspofungins” and “echinocandins” were used synonyms! It is a big mistake.
  5. Generally, between lines 99-141 should be re-written. It is also important to read before re-write the MS : Pappas PG et al. 2016. Clinical practice guideline for the management of candidiasis: Update by the Infectious Diseases Society of America. Clin Infect Dis. 2016;62:e1-50.
  6. Lines 206-208. The lowest cost can be seen in case of amphotericin B deoxycholate.

However, 1 gram cumulative dose of amphotericin B deoxycholate causes acute kidney failure with 40% probability.

  1. Alternative therapies and clinical applicability section: theoretically antifungal combination is a good alternative for the treatment of invasive C. auris infections, practically combinations of antifungals rarely led to improve the outcome in critically ill patients.
  2. With unknown reason(s) I have seen the “Abbreviations” section (lines 222-234)

After extensive work this MS may be acceptable.

Author Response

Answer Letter for “Echinocandins as biotechnological tools for treating Candida aurisinfections”

Reviewer 1 wrote:

Comment 1:Line 29. It is not true that most C. aurisisolates are resistant to the main 3 antifungal classes.

Our answer: The authors partially agreed with the reviewer. The sentence was rewritten and the word “most” was replaced by “several”, as the literature reports a large number of isolates that are already resistant to the main groups of antifungals for clinical use.

Lines 31 – 32:“since several isolates have already been confirmed as resistant to three antifungals drugs classes ”

Comment 2:Line 33. Chronic otitis media or externa?

Our answer:Chronic otitis media. This information was added to the sentence as follows:

Line 36:“chronic otitis media”

Comment 3:Lines 46-47: Again, the Authors considered C. aurisresistant to all antifungal classes.

Our answer: The authors are taking into account that the available literature shows that most C. aurisisolates evaluated are already resistant to commercial drugs. We reorganized the sentence as follows:

Lines 55 - 56: “The major concern about C. aurisinfections is the fact that the vast majority of isolates already identified and evaluated have resistance to the four main antifungals classes:”

Comment 4:Lines 54-55. Although, the Authors wanted to provide a comprehensive update of C. auris infections, they did not mention:

-the five different clades of C. auris with different phenotypic characteristics, virulence and antifungal susceptibility profiles (Szekely et al., J Clin Microbiol. 2019;57(5). pii: e02055-18, Forgács et al., Emerg Microbes Infect. 2020 Dec;9(1):1160-1169. doi: 10.1080/22221751.2020.1771218.).

-risk factors for C. auris infections.

Our answer: As suggested by the reviewer, the information about the clades and risk factors for C. aurisinfections were added to the manuscript as follows:

Lines 27 – 29:“Furthermore, risk factors that aggravate C. aurisinfections include conditions such as diabetes mellitus, cardiovascular surgical interventions and gastrointestinal pathologies, haematological malignancies and even corticosteroid therapy.”

Lines 41 – 45:“Currently, it is possible to separate C. aurisisolates into five different clades, according to their geographical origin (South Asian, East Asian, South African, South American and Iranian). This classification was carried out using whole genome sequence data from global clinical data. Each of these clades differs phenotypically. Moreover, in terms of virulence these isolates follows the order: South American> South Asian> (South African = East Asian).”

Also, the references bellow were added to the manuscript:

Line 29 and 45:Forgacs, L., et al., Comparison of in vivo pathogenicity of four Candida auris clades in a neutropenic bloodstream infection murine model.Emerg Microbes Infect, 2020. 9(1): p. 1160-1169.

Line 45:Szekely, A., A.M. Borman, and E.M. Johnson, Candida auris Isolates of the Southern Asian and South African Lineages Exhibit Different Phenotypic and Antifungal Susceptibility Profiles In Vitro.J Clin Microbiol, 2019. 57(5).

Comment 5:Line 88. “in fragile patients”. What does it mean? Probably, critically ill patients.

Our answer:The authors intended to say hemodynamically unstable patients. The sentence was rewritten accordingly.

Lines 98 - 99:“especially in hemodynamically unstable patients after triazoles treatment.”

Comment 6:Line 102. “deep-tissue candidiasis”. What does it mean?

Our answer: This terminology is associated with infections involving Candidainfecting internal organs, also known as “deep-seated tissue candidiasis” or “deep organ infections”. For better comprehension, the sentence was modified.

Line 112:“deep organ infections”

Comment 7:Line 119. “cavity infections”. What does it mean?

Our answer: In fact, it is not “cavity infections”, one word was missing. It was fixed accordingly, as follows:

Line 129:“oral cavity infections”

Comment 8:Lines 124-133. “Caspofungins” and “echinocandins” were used synonyms! It is a big mistake.

Our answer: The authors partially agree with the reviewer. Our aiming was to show the general behavior of echinocandins against yeast cells and compare to the caspofungins behavior at the same situation. Moreover, as caspofungins are echinocandin members, we believe that the modified sentence is now sound and clear.

Lines 134 -135:Echinocandins, in general, manage to destabilize the integrity of the cell wall and reduce its stiffness, consequently causing cell lysis due to low osmotic pressure.

Comment 9:Generally, between lines 99-141 should be re-written. It is also important to read before re-write the MS: Pappas PG et al. 2016. Clinical practice guideline for the management of candidiasis: Update by the Infectious Diseases Society of America. Clin Infect Dis. 2016;62:e1-50.

Our answer: The section was partially re-written andthe recommended article was added to the manuscript:

Lines 134 – 174:“Echinocandins, in general, manage to destabilize cell wall integrity reducing its stiffness and consequently causing cell lysis due to low osmotic pressure. However, even though they belong to the echinocandin group, caspofungin has not been reported as an effective molecule for C. aurisbiofilms control [25]. Notoriously, caspofungins are considered effective against yeasts that form biofilms. However, they are inactive against C. aurisbiofilms, an unexpected event, since these molecules are normally effective against Candidaspecies biofilms [41,42]. In addition, they are not used as a treatment for urinary infections caused by C. auris, as they fail to reach the required therapeutic concentrations of these compounds in the urine [25]. The survival C. auriscapability in hospital environments may be related to yeast biofilm formation. In this way, Sherry and co-workers [42]tested the C. aurisability to form biofilms and further demonstrated that the species can adhere to polymeric surfaces. In addition their results demonstrated a significant increase in resistance, highlighting that caspofungin, usually effective against Candidabiofilms, was ineffective against planktonic cells and C. aurisbiofilms [42].

Similar to caspofungin, micafungin originated from the cleavage and addition of an N-acylated side chain to the natural hexapeptide derived from C. empetri. Structurally, it is identified as 1 - [(4R, 5R) -4,5-dihydroxy-N2- [4- [5- [4- (pentyloxy) phenyl] -3-24 isoxazolyl] benzoyl] -L-ornithine] - 4 - [(4S) -4-hydroxy-4- [4-hydroxy-3- (sulfooxy) phenyl] -25 L-threonine] monosodium salt[25]. The FDA approved this molecule in March 2005 for use in adults and in 2013 for pediatric treatment. Its use covers the treatment of adult and child patients with esophageal candidiasis and more delicate cases such as hematopoietic cell translation during neutropenia, being considered effective in 86% of the cases [39]. Similarly to the two aforementioned echinocandins, there are already strains of C. aurisresistant to micafungin with MICs ³4 mg.mL-1 [1,4,33].

In addition to the echinocandins already discussed, rezafungin (also named CD101), a new drug in this class, has been developed. It represents a structural anidulafungin analogue and has a hexapeptide with a lipophilic tail and choline moiety at the C5 ornithine position. Modification in this analogue were carried out in order to increase the plasma stability, aqueous solutions and at high temperatures, to avoid hepatotoxicity [43], as well as increasing its half-life (30 to 40 h) and safety profile [44]. According to studies by Berkow & Lokart [45], rezafungin presented better MIC results than did other echinocandins, ranging from 0.03 to 8 mg.mL-1, while towards the same isolates the values presented by micafungin (0.5 to >8 mg.mL-1), caspofungin (0,5 to >16 mg.mL-1) and anidulafungin (1 to >16 mg.mL-1) were lower. Lepak and co-workers [46]demonstrated the potent in vivo rezafungin activity against clinical strains of C. auris. The authors suggested that the clinical dose of 400 mg administered once a week is enough to meet or exceed the pharmacodynamics target for >90% of isolates, being a really useful drug for patients infected with C. auris[46]. Still, some isolates have already been described as rezafungin resistant [43].

Although still in small proportions, C. aurisresistance to lipopeptides is of great concern to researchers. It is also important to note that the resistance of Candidaspecies to echinocandins is uncommon, but has been growing, mainly due to its widespread prophylaxis, being detected predominantly in isolates that already show resistance to azoles, suggesting a possible cross-resistance event [47]. The Infectious Diseases Society of America (IDSA) recommends that azole susceptibility tests should be performed for all blood streams and all possibly relevant Candidaisolates. As well as susceptibility test to echinocandins should be carried out mainly in cases of previous infections by Candida glabrataand Candida parapsilosis [3].

The occurrence of echinocandins resistance in Candidaspecies was first reported in 2005, where strains with mutations in the FKSgenes of resistant C. albicans(FKS1) and C. glabrata(FKS2) presented low sensitivity to caspofungin [25]. Recent reports indicate that ~2% of C. auriscould be resistant to echinocandins [2,25]. C. auristolerance to echinocandins was also in vitroreported, revealing that some isolates showed high tolerance to anidulafungin, micafungin and caspofungin [33,37].”

The following reference was added:

Pappas, P.G., et al., Clinical Practice Guideline for the Management of Candidiasis: 2016 Update by the Infectious Diseases Society of America.Clin Infect Dis, 2016. 62(4): p. e1-50.

Comment 10:Lines 206-208. The lowest cost can be seen in case of amphotericin B deoxycholate.

However, 1 gram cumulative dose of amphotericin B deoxycholate causes acute kidney failure with 40% probability.

Our answer:The authors understand the reviewer’s concern and the sentence was rewritten.

Lines 222 – 227:“Treatments with micafungin and caspofungin, intravenously administered, represent a low cost when compared to conventional drugs such as amphotericin B. Also, when dealing specifically with the control of infections caused by C. auris, the use of combined therapies proved to be more promising than the use of echinocandins alone. However, it is necessary to be attentive due to prolonged use of these molecules may trigger kidney problems (specially when combined with drugs as amphotericin B deoxycholate) and even resistance, as previously discussed.”

Comment 11:Alternative therapies and clinical applicability section: theoretically antifungal combination is a good alternative for the treatment of invasive C. aurisinfections, practically combinations of antifungals rarely led to improve the outcome in critically ill patients.

Our answer: The authors understand reviewer’s concern. However, here our aim is to show the reader the possibilities that are being explored by researchers in the field of echinocandins andC. auris. Since the anti-C. aurismolecules discussed here are available and in use for the treatment of candidiasis, we believe it is of great value to provide the readers with the alternatives of synergism between them, even in initial tests.

Lines 227 – 229:“Furthermore, it is important to note that the tests regarding combination therapies were performed in in vitroassays, and that it is essential to carry out in vivotests, considering toxicity analyzes and adequate concentrations for therapeutic application, mainly for patients in critical health conditions.”

Comment 12:With unknown reason(s) I have seen the “Abbreviations” section (lines 222-234)

 Our answer:The authors followed the formatting recommendations provided by the journal. Therefore, this section was not removed from the updates manuscript.

Reviewer 2 Report

The manuscript entitled "Echinocandins as biotechnological tools for treating Candida auris infections" nicely guides the reader into the field of Candida auris infections and how echinocandins can help to treat these infections. The review comprehensively summarizes the field and gives a detailed view on the class of lipopeptides. Interesting is the table listing (variants) of echinocandins (table1). I like figure 1 (the global map) listing the time course of C. auris infections since 2009, this is a very nice way to summarize the current status of the global progress of this infection. The literature cited (59 publications) is very comprehensive and allows to go deeper into the subject also for readers who are not too familiar with C. auris and/or echinocandins. This in my opinion is exactly one task of a good review. I suggest to accept this manuscript for publication. 

Author Response

The manuscript entitled "Echinocandins as biotechnological tools for treating Candida auris infections" nicely guides the reader into the field of Candida auris infections and how echinocandins can help to treat these infections. The review comprehensively summarizes the field and gives a detailed view on the class of lipopeptides. Interesting is the table listing (variants) of echinocandins (table1). I like figure 1 (the global map) listing the time course of C. auris infections since 2009, this is a very nice way to summarize the current status of the global progress of this infection. The literature cited (59 publications) is very comprehensive and allows to go deeper into the subject also for readers who are not too familiar with C. auris and/or echinocandins. This in my opinion is exactly one task of a good review. I suggest to accept this manuscript for publication. 

Our answer: The authors are grateful for the reviewer’s comments. Here, we aimed at organizing a straightforward review article that would be of great interest in the fields of drug design and C. auris infections. We are glad to know that our manuscript caused a positive impact. 

Round 2

Reviewer 1 Report

The changes looks good.